# Clinical Trial Data: Both Parents Having Psychiatric Symptoms as Risk Factor for Children’s Mental Illness

**DOI:** 10.3390/children9111697

**Published:** 2022-11-05

**Authors:** Hannah Suess, Silke Wiegand-Grefe, Bonnie Adema, Anne Daubmann, Reinhold Kilian, Antonia Zapf, Sibylle M. Winter, Martin Lambert, Karl Wegscheider, Mareike Busmann

**Affiliations:** 1Department of Child and Adolescent Psychiatry, Psychotherapy and Psychosomatics, University Medical Center Hamburg-Eppendorf, 20251 Hamburg, Germany; 2Department of Medical Biometry and Epidemiology, University Medical Center Hamburg-Eppendorf, 20251 Hamburg, Germany; 3Department of Psychiatry and Psychotherapy II, Ulm University at Bezirkskrankenhaus Guenzburg, 89312 Guenzburg, Germany; 4Department of Child and Adolescent Psychiatry, Psychosomatics and Psychotherapy, Charité—Universitätsmedizin Berlin, 13353 Berlin, Germany; 5Department of Adult Psychiatry, Psychotherapy and Psychosomatics, University Medical Center Hamburg-Eppendorf, 20251 Hamburg, Germany

**Keywords:** children of mentally ill parents, children’s mental illness, parental mental illness, parental personality disorder

## Abstract

Children of mentally ill parents represent a particularly vulnerable risk group for the development of mental illness. This study examines whether there is a predictive association between children’s psychiatric symptomatology and (1) the clinical diagnosis according to the International Statistical Classification of Diseases and Related Health Problems (ICD-10) of their mentally ill parent as well as (2) to families both parents showing psychiatric symptoms. The study is part of the multicenter controlled trial project “Children of Mentally Ill Parents” (CHIMPS). For this purpose, the psychiatric symptomatology of the mentally ill parent (N = 196) and his or her partner (N = 134) as well as the psychiatric symptomatology of their children aged 4 to 18 years (N = 290) was measured using clinical rated ICD-10-diagnosis, self-rated Brief Symptom Inventory (BSI), and Child Behavior Checklist (CBCL). Using multilevel analyses, the severity of the parental psychiatric symptomatology (BSI) was identified as a significant predictor of children’s psychiatric symptomatology (CBCL). Children of parents with a personality disorder (ICD-10) were not more affected than children of parents with another ICD-10-diagnosis. However, children with two parents showing psychiatric symptoms (CBCL) were significantly more affected than children with one mentally ill parent. The results of this study support the well-known view that parental mental illness is a risk factor for children’s psychiatric symptoms. Therefore, increased support, especially in high-risk families, both parents having psychiatric symptoms, is highly necessary and should be implemented in the future psychotherapeutic family care.

## 1. Introduction

Due to their increased risk of mental illness, children of mentally ill parents are a particularly vulnerable risk group [1]. According to the World Health Organization (WHO), mental health is more than the absence of mental disorders and is defined as a state of mental well-being. However, mental disorders are characterized by a clinically significant disturbance and are described in diagnostic systems such as the International Statistical Classification of Diseases and Related Health Problems (ICD-10) [2]. In 2008, researchers concluded that in German-speaking countries, an average of 30% of the inpatient psychiatric population are parents of underage children [3]. On average, individuals with mental disorder have children as often as their healthy reference population [4]. In one year, approximately three million children in Germany are therefore confronted with the mental illness of a parent [5]. There is a wide range of studies that confirm that children of mentally ill parents have an increased risk of developing a mental illness themselves and therefore comprise a particularly important risk group [1,6]. Three out of four mental disorders develop during childhood or adolescence [7].

Beardslee, Keller, Lavori, Staley and Sacks (1993) [8] found that depression and other parental affective disorders are often related to severe affective disorders in their children. In recent studies, children of parents with a depressive disorder showed a two to four times higher risk of also developing a depression compared to children with healthy parents [3,4,9]. In a meta-analysis of 193 studies, significantly higher levels of internalizing and externalizing behavior, as well as general psychopathology and negative affect and behavior, were found in children of depressed mothers [10]. In a review, Lawrence, Murayama and Creswell (2019) [11] also found an increased risk in children of parents with an anxiety disorder of developing an anxiety disorder or depression. These results relate in particular to the parental diagnoses of generalized anxiety disorder and panic disorder. Children showed increased symptoms of generalized anxiety disorder, separation anxiety and specific phobias [11]. Children of parents with eating disorders also show emotional difficulties and behavioral problems, such as hyperactivity and difficulties in interacting with peers. In a study of Scandinavian families with parental eating disorder, both internalizing and externalizing abnormalities were found in the child participants [12]. Moreover, studies have been shown that children of parents with post-traumatic stress disorder showed a higher number of internalizing disorders such as depression and anxiety as well as externalizing dysfunctional behavior such as aggression [13].

Compared to children of parents with an axis-one disorder of the Diagnostic and Statistical Manual of Mental Disorders IV [14], children of parents with a personality disorder (PD) were found in several studies to be a particularly affected population with an increased risk of mental health difficulties [4,6,15]. Up to 80% of children of parents with a borderline personality disorder show disorganized parent-child attachment [16]. This suggests a rather negative trajectory in the development of these children [3]. Mothers with borderline personality disorder often demonstrate low sensitivity, high intrusiveness and greater difficulty in recognizing their children’s emotions and reacting to them adequately [15]. Furthermore, parent-child constellations often contain a lack of clarity about the distribution of roles within the family [17]. Affected parents typically change between an overprotective / dismissive and hostile parenting style, and experience a high level of parenting stress [18]. Furthermore, parents with borderline personality disorder often have comorbid conditions, such as affective disorders, anxiety disorders, substance abuse, and eating disorders, which in many cases can further increase the severity of stress [15,19]. In a study by Apter et al. (2017) [20], affected two-month-old infants already showed differences in their visual behavior and willingness to interact compared to a control group with healthy mothers. Older children (mean 11 years) of mothers with borderline personality disorder often have poorer interactions with their mother and a wide range of cognitive and / or behavioral problems [15]. Mental disorders frequently found in the affected children were conduct disorders, hyperkinetic disorders, symptoms of borderline personality disorder, separation anxiety disorder, self-destructive behavior and dissociation [21].

Studies have shown that an important mediator between the mental illness of parents and the mental illness of their children is dysfunctional parenting style and parent-child interactions, where the parental sensitivity is a particularly important factor [5,22]. For example, hostile behavior of parents is more common among mentally ill parents [5]. Moreover, there is evidence that maladaptive emotion regulation strategies of parents with depression are related to their children’s internalizing symptoms [23]. Loechner and colleagues (2020) [24] identified in their clinical research an increasingly passive emotion regulation style in four to seven year old children of mothers with clinical depression. Thus, increased difficulties in emotion regulation are a phenomenon often seen in children of mentally ill parents [12,13,21,24]. Maladaptive emotion regulation strategies are among the most common risk factors for the development of psychological difficulties [25]. While parental mental illness in itself can evoke a maladaptive development in children, linked consequences further promote their children’s vulnerability for mental problems [1]. This includes financial problems and interpersonal difficulties, which include marital difficulties and social isolation, as well as discrimination from society [1,26]. Furthermore, the impact of a parental mental illness varies according to the type, severity and chronicity of the illness [1]. In particular, severe chronic parental depression, post-traumatic stress disorder and borderline personality disorder can lead to disorganized attachment style in children [16]. This form of attachment is itself a risk factor for child development [16]. Frequently, a “fright without solution” phenomenon can be observed in these children [16]. They feel rejected, left alone, increasingly insecure, and cannot rely on their parent to regulate those intense emotions [27].

However, this negative expectation towards attachment figures can be compensated by significant others, for example, a second healthy parent [16]. A study of Chang, Halpern and Kaufman (2007) [28] found that the presence of a healthy parent who is supportive and caring towards the child can work as a buffer against negative child development outcomes. Moreover, supportive relationships outside the family can strengthen the resilience of children with mentally ill parents [29]. Studies have found positive effects when children had access to stable, non-family attachment figures. These included teachers and other educational staff, as well as family friends [30]. Therefore, the children’s social network is an important protective factor for their mental health [31]. In addition, for parents, having emotional supportive relationships can help them be more in control of their parenting skills, which can help to break the cycle of transgenerational psychopathology [32]. Because a mental illness becomes more resistant the longer it remains untreated, it is highly important to detect mental health issues in an early stage [7]. For the prevention of mental illness as well as for the treatment of mentally ill children, it is therefore important to study the association of parental mental illness on the health of their children in order to understand the underlying mechanisms and factors involved. 

Based on the research literature reviewed above, the following research questions were tested to analyze the association between the psychiatric symptomatology of mentally ill parents and the psychiatric symptomatology of their children: (1) Is there a relationship between the psychiatric symptomatology of the mentally ill parents and the psychiatric symptomatology of their children? (2) Are children of parents with a PD according to ICD-10 more affected by psychiatric symptoms than children of parents with another ICD-10 diagnosis? (3) Are children with both parents having psychiatric symptoms more affected by psychiatric symptoms than children with one mentally ill and one mentally healthy parent? It is hypothesized that there is an association between the psychiatric symptoms of the mentally ill parent and those of the child. Further, it is assumed that children of parents with a PD diagnosis are more affected than parents with another psychiatric diagnosis and that children of both parents having psychiatric symptoms are more affected by psychiatric symptoms than children with only one mentally ill parent. 

## 2. Materials and Methods

### 2.1. Participants

The cross-sectional survey is part of the multicenter controlled trial study “Children of Mentally Ill Parents” (CHIMPS) at the University Medical Centre Hamburg-Eppendorf from 2014 to 2017 [33]. In addition to the University Medical Centre Hamburg Eppendorf, other recruitment centers were the University Medical Centre Leipzig, the Department of Psychiatry and Psychotherapy II of the Ulm University at the Bezirkskrankenhaus Günzburg, the Vitos Klinik Rheinhöhe in Wiesbaden-Rheingau, the Landschaftsverband Westfalen-Lippe clinic Gütersloh, the Charité in Berlin, and the Canton Hospital Winterthur, Switzerland. The sample consisted of N = 216 families. Inclusion criterion of the CHIMPS project was a parental psychiatric diagnosis according to the ICD-10 [34]. The ICD-10 diagnosis was given by the treating psychiatrist. Furthermore, inclusion criteria were at least one underage child that is living in the household or is in regular contact with the parent, agreement to participate in the study, and sufficient knowledge of the German language of both parents and children. Exclusion criteria of the CHIMPS project were children and parents with severe suicidal tendencies or acute psychotic symptoms for whom outpatient treatment would not be sufficient. The ethics committee of the medical association in Hamburg, Germany approved the study.

For this survey, due to complete missing data, two families (*n* = 2) had to be excluded from the data set. Seven families (*n* = 7) were excluded from the data set due to missing data. Eleven families (*n* = 11) had to be excluded because the children were under the age of four years or over the age of 18 years, resulting in a sample size of N = 196 families with a total of *n* = 290 children. N = 69 of the families have an only child, *n* = 84 have two children, and *n* = 43 have three to five children. In *n* = 131 families, there was a partner in addition to the mentally ill parent. Consequently, *n* = 62 families were single-parent families. 

The age of the mentally ill parent (N = 196) ranged between 23 and 57 years, with an average of 40.3 years (SD = 6.96). The sample consists of mainly female mentally ill parents (*n* = 147)). N = 109 of the mentally ill parents are married, *n* = 34 divorced, *n* = 45 single, and *n* = 5 widowed. 

The average age of the partners (*n* = 134) is 40.7 years, with an age range between 26 and 59 years (SD = 6.5). About one third of the partners (*n* = 50) are female and two thirds (*n* = 84) are male. N = 98 of them are currently married, *n* = 15 divorced, and *n* = 21 singles. 

Of the children (*n* = 290), *n* = 152 are female and *n* = 138 are male with an age range from 4 to 18 years and an average age of 10.0 years (SD = 4.03).

### 2.2. Measures

#### 2.2.1. Brief Symptom Inventory (BSI)

In order to determine the parental psychiatric symptomatology, the parents received the self-report questionnaire Brief Symptom Inventory (BSI) [35]. On a five-point Likert scale, participants rate their subjective impairment on 53 physical and psychological symptoms (53 items). A main scale of the BSI is the General Severity Index (GSI), which is used in this study to assess the psychiatric symptoms on an individual and is calculated by all 53 items [27]. Reliability tests revealed a wide range of internal consistency values between Cronbach’s α = 0.39 and α = 0.96 [36]. For the current study, sample reliability tests revealed a high internal consistency value of Cronbach’s α = 0.97. 

#### 2.2.2. Child Behavior Checklist (CBCL)

In order to assess children’s psychiatric symptomatology, the German version of the Child Behavior Checklist (CBCL/4–18) was used to measure behavioral, emotional and somatic difficulties, as well as social skills of children aged 4 to 18 years [37]. The CBCL collects the parents’ judgement of their children from the past six months [38]. The items are measured on a three-level scale with the response categories (0) “not applicable”, (1) “somewhat or sometimes applicable”, and (2) “exactly or frequently applicable”. The CBCL consists of 118 Items. The evaluation of the method used results of three scales: the internalizing (CBCL-I, 31 items), externalizing (CBCL-E, 33 items), and total problem score (CBCL-T, 118 items) [29]. Achenbach [29] defined the 90th percentile (T > 63) as the cut-off value for the main scales CBCL-I, CBCL-E and CBCL-T. According to this, a child is considered to be conspicuous if it is rated more conspicuous on a scale than 98% of its peers. The German version of the CBCL revealed good internal consistency values of Cronbach’s α > 0.80 for the superior scales CBCL-I and CBCL-E in clinical and field samples [38]. In this study, the internal consistency of the CBCL-T is satisfying with Cronbach’s α = 0.84. The CBCL-I (Cronbach’s α = 0.90) and CBCL-E (Cronbach’s α = 0.92) scales also had high reliability values.

### 2.3. Study Design and Procedure

The multicenter CHIMPS project is a randomized controlled trial with four measurements. For this study, the first measurement was used that represents the baseline survey and took place before randomization and implementation of an intervention. For this study, the ICD-10 diagnoses, BSI, and CBCL were applied. The ICD-10 diagnoses were given by the attending psychiatrist or psychotherapist and were transmitted to the project. The self-report questionnaires (BSI, CBCL) were fulfilled by the families. Moreover, the self-report questionnaires contained a section with questions on socio-demographic data on parents and children. Before participating in the project, the families were informed about the project and filled out the information consent.

### 2.4. Statistical Analysis

Based on the nested data structure, using a multilevel analysis a multiple random intercept model was calculated [39]. Three models were calculated with each a different criterion to analyze (CBCL-T, CBCL-I, CBCL-E). For each model on level one (children level, L1), age, and sex of the children were included as predictors. On level two (family level, L2), the psychiatric symptomatology (BSI GSI) as well as the psychiatric diagnoses (ICD-10 diagnosis) of the mentally ill parent, their age, and sex were included as predictors.

To investigate the first research question, the L2 predictor variable of the psychiatric symptomatology (BSI GSI) of the mentally il parent was centered using the grand mean centering before it was included in the analysis, so it can be interpreted correctly [31]. Initially, a null model was computed without any predictors to analyze the effect of family affiliation. Secondly, the parental L2 predictor BSI GSI was added. Finally, the control variables age and sex of both the parent (L2), and child (L1) were added to the model. 

For the second research question, the L2 predictor diagnosis of the mentally ill parent (ICD-10 diagnosis) was divided into two groups: ICD-10 PD disorder and another ICD-10 disorder. Again, a null model was computed without any predictors. Secondly, the parental L2 predictor ICD-10 diagnosis was added. Lastly, the control variables age and sex of both the parent (L2), and child (L1) were added to the model.

Regarding the third research question, the parental L2 predictor psychiatric symptoms of the partner (BSI GSI) were divided into two groups using the cut-off value T ≥ 63, which was recommended by the authors of the BSI [27]. Participants who achieved a T-value higher or equal to 63 are therefore considered clinically relevant cases. Again, a null model was computed without any predictors. Secondly, the parental L2 predictor psychiatric symptoms of the partner (BSI GSI cut-off) were added. Finally, the control variables age and sex of both the parent (L2), and child (L1) were added to the model. 

The analysis was conducted by using the IBM SPSS Statistics 25 statistical software 25.0.

## 3. Results

### 3.1. Descriptive Statistics 

The distribution of the clinical diagnoses according to the ICD-10 of the mentally ill parent and the partner can be seen in Table 1. 83.2% of the mentally ill parents scored above the cut-off value for noticeable psychological stress on the GSI scale of the BSI. In comparison, only 26.6% of the partners reached the cut-off value of the GSI scale of the BSI for noticeable psychological stress. Table 2 shows the outcome of parental psychiatric symptomatology using the GSI of the BSI. On average, the severity of symptoms of the mentally ill parents was M = 1.33 (SD = 0.69) and their partners average severity of symptoms was M = 0.51 (SD = 0.54). Referring to the CBCL-T, 52% of the children reached the cut-off value for clinical significance when rated by the mentally ill parent and 38 % in the rating done by the partner. Table 3 shows children’s psychiatric symptomatology rated by both of their parents using the CBCL, which provides three global scales of CBCL-T, CBCL-I, and CBCL-E.

### 3.2. Mentally Ill Parent’s Psychiatric Symptomatology (BSI GSI) and Children’s Psychiatric Symptomatology (CBCL)

Regarding the CBCL-T the final model with all added predictors (L1: child age, child sex; L2: BSI GSI, parent age, parent sex), family correlation, which represents similar levels of psychiatric symptomatology among siblings in one family, explained 51% of the outcome variance (interclass correlation coefficient: ICC = 0.51). The model consistently improved in all steps, adding predictors as shown in Table 4. In the final model, the CBCL-T scores of children were significantly predicted by BSI GSI (b = 5.4, *p* = 0.00), as well as the parental age (b = −0.3; *p* = 0.01). 

The same model was also tested using the CBCL-I as the criterion. Regarding the final model with all added predictors (L1: child age, child sex; L2: BSI GSI, parent age, parent sex), family correlation explained 44.87% of the outcome variance (ICC = 0.45). Again, the model consistently improved in all steps adding predictors, as shown in Table 4. In the final model, children’s CBCL-I scores were significantly predicted by the BSI GSI (b = 5.7, *p* = 0.00), the parental age (b = −0.3; *p* = 0.02), as well as the children’s age (b = 0.6, *p* = 0.00) and sex (b = 2.8, *p* = 0.02). 

Analyzing the CBCL-E the final model with all added predictors (L1: child age, child sex; L2: BSI GSI, parent age, parent sex) family correlation explained 25.05% of the outcome variance (ICC = 0.25). The model again consistently improved in all steps, adding predictors as shown in Table 4. In the final model, children’s CBCL-E scores were significantly predicted by the BSI GSI (b = 3.9, *p* = 0.00), the parental age (b = −0.2; *p* = 0.03), as well as the children’s age (b = −0.4, *p* = 0.02). The detailed prediction analyses for the first research question are shown in Table 4.

### 3.3. Mentally Ill Parent’s Diagnosis (ICD-10) and Children’s Psychiatric Symptomatology (CBCL)

The analysis regarding children’s psychiatric symptomatology of parents with a PD did not show significant results. The inclusion of the L2 predictor ICD-10 diagnosis of the mentally ill parent did not improve any model (CBCL-T, CBCL-I, CBCL-T) significantly (model 1; *p* > 0.05). Only the inclusion of the control variables (parent’s and children’s age and sex) significantly improved the model (model 2; *p* < 0.01). Only parental age (b = −0.3, *p* = 0.01) was a significant predictor for children’s CBCL-T. Regarding children’s CBCL-I scores, the children’s age (b = 0.6, *p* = 0.00), children’s sex (b = 2.9; *p* = 0.02) as well as the parental age (b = −0.3; *p* = 0.02) were found to be significant predictors. For children’s CBCL-E scores, the children’s age (b = −0.4, *p* = 0.02) as well as the parental age (b = −0.3; *p* = 0.03) were significant predictors. The detailed prediction analyses for the second research question are shown in Table 5.

### 3.4. Mentally Ill Parent’s and Partner’s Psychiatric Symptomatology (BSI GSI Cut-Off) and Children’s Psychiatric Symptomatology (CBCL)

Regarding the CBCL-T the final model with all added predictors (L1: child age, child sex; L2: BSI GSI cut-off, parent age, parent sex), family correlation explained 54.21% of the outcome variance (ICC = 0.54). The model consistently improved in all steps, adding predictors, as shown in Table 6. In the final model, a second mentally ill parent (BSI GSI cut-off; b = 7.7, *p* = 0.00), as well as the parental age (b = −0.3, *p* = 0.02) were significant predictors of children’s CBCL-T scores.

The same model was also tested using the criterion CBCL-I. Regarding the final model with all added predictors (L1: child age, child sex; L2: BSI GSI cut-off, parent age, parent sex), family correlation explained 47% of the outcome variance (ICC = 0.47). The model consistently improved in all steps adding predictors as shown in Table 6. A second mentally ill parent (BSI GSI cut-off) was found to be a significant predictor for children’s CBCL-I scores (b = 8.2, *p* = 0.00). In addition, the children’s age was a significant predictor for children’s CBCL-I scores (b = 0.6, *p* = 0.00). 

Regarding the CBCL-E the final model with all added predictors (L1: child age, child sex; L2: BSI GSI cut-off, parent age, parent sex), family correlation explained 26.47% of the outcome variance (intraclass correlation coefficient (ICC) = 0.26). As in previous analyses, the model consistently improved in all steps, adding predictors as shown in Table 6. Again, a second mentally ill parent (BSI GSI cut off) was found to be a significant predictor for children’s CBCL-E scores (b = 4.8, *p* = 0.01). Furthermore, the parental age was a significant predictor for children’s CBCL-E scores (b = −0.4, *p* = 0.02). The detailed prediction analyses for the third research question are shown in Table 6.

## 4. Discussion

The aim of the study was to analyses relationships between the psychiatric symptomatology of mentally ill parents and the psychiatric symptomatology of their children. Furthermore, the study investigated differences in children’s psychiatric symptomatology among parents with different ICD-10 diagnoses as well as the association of a second mentally ill parent.

The results regarding the first research question show that the severity of parents’ psychiatric symptomatology was a significant predictor for the psychiatric symptomatology of their children. The CGI of the BSI was a significant predictor for all three scales of the CBCL (CBCL-T, CBCL-I, CBCL-E). The more a mentally ill parent was affected by their psychiatric symptomatology, the higher the children’s ratings of behavioral and emotional problems were. Another significant predictor was the parental age. A young parental age was a significant predictor for poorer CBCL scores. Furthermore, it was found that young children tended to exhibit externalizing symptoms, while older children showed more internalizing symptoms. The analyses concerning the second research question did not provide significant results regarding the predictive quality of parental PD for children’s psychiatric symptomatology. An ICD-10 PD diagnosis was no significant predictor for any of the three CBCL scales (CBCL-T, CBCL-I, CBCL-E), as it was compared to another ICD-10 diagnoses in the current study. The results of the third research question show that children with two mentally ill parents were significantly more affected than children with one mentally ill and one healthy parent. Children’s CBCL scores on all three scales (CBCL-T, CBCL-I, CBCL-E) were significantly higher when the Partner reached the cut-off value for clinical significance in the GSI of the BSI. 

Consistent with the literature, high psychiatric symptom severity of mentally ill parents is a risk factor for children’s psychiatric symptoms [1,16,40]. Research suggests that the type of parental diagnosis is less important for child development than the chronicity and severity of the parental mental illness [41]. Abela, Skitch, Auerbach, and Adams (2005) [42] found that children of parents with comorbid mental disorders showed higher psychopathological symptoms than children of parents with a single diagnosis. Several studies report a positive association between the severity of a parent’s depression and the frequency of disorganized attachment in their children [43,44]. Disorganized attachment style is one of the main risk factors for later psychopathology [16]. Furthermore, severe parental mental illness often leads to a long-term separation of the children from the parent and to a splitting of the family, which creates additional stress [41]. The results of the present study confirm the view that symptom severity plays a decisive role, which points to the importance of supporting parents with severe psychiatric symptoms and their children as a means to support positive child development.

In contrast to the findings of the present study, previous studies found that children of parents with a PD disorder were significantly more affected than children of parents with another psychiatric disorder [6,45,46]. This could be explained by the following: The parental diagnosis is based on reports by the attending psychiatrist or psychotherapist. Therefore, mentally ill parents who participated in the study could have had an undetected PD (a false negative) and could have been assigned to the wrong group (another ICD-10 diagnosis) due to comorbid diagnoses. Further, the assessment of children’s psychiatric symptoms can be biased by the mental illness of the parent. Especially parents with borderline personality disorder often have difficulties in recognizing their children’s emotions [16]. Fonagy and Bateman [44] report problems in the mirroring of affects in people with borderline personality disorder. The affected persons fluctuate between blocking of meaning and incorrect mentalization [45]. Another possibility is, that the number of parents with a PD may have been too small (27.5%), making it difficult to achieve statistical significance for a given effect. 

Regarding the third research question, the results show that the number of mentally ill parents in the family is a significant predictor for children’s psychiatric symptomatology. It is possible that a mentally healthy parent promotes children’s resilience by compensating risks caused by the mentally ill parent. Some research indicates that the psychological status of the second parent has a significant influence on the child’s development [16,41]. The presence of a healthy parent who is supportive and caring towards the child can work as a buffer against negative child development outcomes [28]. A study of Wilson, Bobier and Macdonald (2004) [47] found that mentally ill parents with negatively affected relationships and/or the absence of a partner are more likely to have children with poorer mental health outcomes. Furthermore, children who grow up with only one mentally ill parent are at greater risk than children where the healthy partner lives in the household [1]. Moreover, for mentally ill parents, having an emotional supportive relationship can help them be more in control of their parenting skills which also promotes child development [32].

As already describe above, a limitation of this study is the assessment of children’s psychiatric symptomatology, which was based only on the parental reports. The literature confirms that parents who experience higher levels of symptoms also rate their children as having higher internalizing or externalizing symptoms [48,49]. An additional external psychological assessment would be a more valid way to evaluate children’s psychiatric symptomatology. Further, an assessment of the clinical parental diagnosis using the SCID I and SCID II would be useful in future research [50,51]. Moreover, there are new concepts in the Diagnostic and Statistical Manual of Mental Disorders 5 (DSM-5; American Psychiatric Association, 2013) and the 11th edition of the World Health Organization’s International Classification of Diseases (ICD-11), such as the alternative model for PD diagnosis of the DSM-5 [52,53].

The results of the study show how important a mentally healthy caregiver or contact person in the life of a child with mentally ill parents is to compensate the risks caused by the parental mental illness. Therefore, additional support for these high-risk families with two mentally ill parents is necessary. The findings point to certain characteristics that indicate an increased psychopathological risk in children of mentally ill parents. Increased parental symptom severity is an increased risk for children’s psychiatric symptoms. Based on this information, high-risk families can be identified early on, and preventive interventions can be implemented at an early stage to promote improved child development. In order to provide sustainable help for the difficulties experienced by children of mentally ill parents, there is a need for multidisciplinary child-, parent- and family-related programs, such as counselling services, outpatient group programs, special outpatient facilities for mentally ill parents, as well as inpatient and day-care treatments [54]. Assisted housing projects for affected families or sponsorship models for children are also a potential support option to strengthen the children’s resilience [54]. For high-risk families with two mentally ill parents in particular, assisted living projects or sponsorship models could be a helpful way to integrate a mentally healthy significant other into the child’s life.

## 5. Conclusions

Overall, the results of this study support the view that parental mental illness is a risk factor for child development. Furthermore, the extent of the risk depends on the psychiatric symptom severity of the mentally ill parent. Children raised by a single mentally ill parent or two parents with mental illness are at the highest risk. Therefore, it is important to implement additional support for these high-risk families to prevent child mental disorders. Children’s resilience can be strengthened by establishing relationships with mentally healthy attachment figures who play an emotionally supportive role in the life of a child. 

## Figures and Tables

**Table 1 children-09-01697-t001:** Distribution of ICD-10 diagnoses among mentally ill parents.

ICD-10 Diagnosis	N	%
Personality Disorders	56	25.9
Psychological and Behavioral Disorders caused by Psychotropic Substances	15	6.9
Schizophrenia, Schizotypal and Delusional Disorders	12	5.6
Affective Disorders	164	75.9
Anxiety Disorders and Reactions to Severe Stress	55	25.5
Eating Disorders	13	6.0
Intellectual Disabilities	1	0.5
Developmental Disorders	1	0.5
ADHD and other Behavioural and Emotional Disorders beginning in childhood and adolescence	8	3.7

Note: N = 196 mentally ill parents; Due to comorbidities, cumulative frequencies are above 100%; ADHD: Attention deficit hyperactivity disorder; ICD: International Statistical Classification of Diseases and Related Health Problems.

**Table 2 children-09-01697-t002:** GSI-BSI of parental psychiatric symptomatology.

	N	Minimum	Maximum	M	SD
Mentally ill parent	196	0	0.02	3.13	1.34
Partner	133	1	0.00	2.62	0.51

Note: BSI: Brief Symptom Inventory. GSI: Global Severity Index.

**Table 3 children-09-01697-t003:** Children’s psychiatric symptomatology reported by the mentally ill parent as well as their partners using the CBCL questionnaire.

		N	Minimum	Maximum	M	SD
Mentally ill parent	CBCL-I	289	1	36.00	91.00	62.80
	CBCL-E	290	0	35.00	88.00	58.80
	CBCL-T	290	0	31.00	85.00	62.00
Partner	CBCL-I	201	2	36.00	97.00	59.33
	CBCL-E	202	1	35.00	89.00	56.49
	CBCL-T	202	1	35.00	97.00	59.15

Note: CBCL: Child Behavior Checklist; CBCL-I: CBCL internalizing scale; CBCL-E: CBCL externalizing scale; CBCL-T: CBCL total problem score.

**Table 4 children-09-01697-t004:** BSI GSI (mentally ill parent) as a predictor for CBCL (children’s psychiatric symptoms).

	Model 0						Model 1						Model 2					
	CBCL-T	CI-95%	CBCL-I	CI-95%	CBCL-E	CI-95%	CBCL-T	CI-95%	CBCL-I	CI-95%	CBCL-E	CI-95%	CBCL-T	CI-95%	CBCL-I	CI-95%	CBCL-E	CI-95%
Fixed Effects																		
Intercept	62.4 **	60.98; 63.82	63.0 **	61.55; 64.45	59.0 **	57.65; 60.35	62.3 **	60.97; 63.61	62.9 **	61.58; 64.27	58.9 **	57.64; 60.24	72.1 **	63.79; 80.41	65.0 **	56.37; 73.61	73.4 **	65.20; 81.67
L1 (Children Level)																		
child age													0.2	−0.09; 0.50	0.6 **	0.32; 0.94	−0.4 *	−0.71; −0.06
child sex													1.8	−0.28; 3.90	2.8 *	0.55; 5.03	0.5	−1.80; 2.78
L1 (Parent Level)																		
parent BSI GSI							5.6 **	3.63; 7.47	5.7 **	3.78; 7.70	4.2 **	2.31; 6.07	5.4 **	3.52; 7.24	5.7 **	3.81; 7.67	3.9 **	2.08; 5.68
parent age													−0.3 **	−0.52; −0.09	−0.3 *	−0.49; −0.04	−0.2 *	−0.46; −0.03
parent sex													−1.7	−4.87; 1.43	−1.3	−4.51; 1.99	−1.2	−4.26; 1.83
Random Effect																		
L1 (child)	47.7 **	36.16; 63.03	66.0 **	50.03; 87.00	76.8 **	58.64; 100.5	46.3 **	35.36; 60.59	63.8 **	48.88; 83.29	74.2 **	57.14; 96.27	45.5 **	34.71; 59.73	55.7 **	42.31; 73.46	72.7 **	55.51; 104.3
L2 (family)	65.4 **	46.79; 91.43	56.7 **	37.08; 86.78	36.4 **	19.74; 67.08	53.2 **	37.45; 75.57	44.3 **	27.90; 70.25	31.3 **	16.60; 59.15	47.4 **	32.61; 68.91	45.4 **	29.26; 70.34	24.2 **	9.90; 65.32
ICC	0.5781		0.4623		0.3216		0.5347		0.4096		0.2970		0.5100		0.4487		0.2505	
−2 Log-Likelihood	2153.4		2185.9		2182.9		2122.9		2154.8		2164.4		2091.9		2116.6		2127.8	
$\chi^{2}/\mathrm{df}$							30.5/1		31.1/1		18.6/1		31.0/4		38.2/4		36.6/4	
*p*							<0.01		<0.01		<0.01		<0.01		<0.01		<0.01	
BIC	2170.4		2202.9		2199.9		2145.6		2177.5		2187.0		2137.2		2161.9		2173.1	

Note: Multilevel-model; N = 290 children nested in 196 families. CBCL-T: CBCL total problem score, CBCL-I: CBCL internalizing scale; CBCL-E: CBCL externalizing scale; BSI GSI: Brief Symptom Inventory Global Severity Index; ICC: Intraclass correlation coefficient; BIC: Bayesian Information Criterion; * *p* ≤ 0.05; ** *p* ≤ 0.01.

**Table 5 children-09-01697-t005:** Parental PD (ICD-10) as a predictor for CBCL (children’s psychiatric symptoms).

	Model 0						Model 1						Model 2					
	CBCL-T	CI-95%	CBCL-I	CI-95%	CBCL-E	CI-95%	CBCL-T	CI-95%	CBCL-I	CI-95%	CBCL-E	CI-95%	CBCL-T	CI-95%	CBCL-I	CI-95%	CBCL-E	CI-95%
Fixed Effects																		
Intercept	62.4 **	60.98; 63.82	63.0 **	61.55; 64.45	59.0 **	57.65; 60.35	62.3 **	60.67; 63.97	62.8 **	61.07; 64.44	58.8 **	57.25; 60.39	74.2 **	65.07; 83.29	66.4	56.93; 75.95	74.7 **	65.96; 83.43
L1 (Children Level)																		
child age													−0.2	−0.10; 0.51	0.6 **	0.31; 0.96	−0.4 *	−0.73; −0.07
child sex													1.8	0.40; 3.97	2.9 *	0.54; 5.23	0.5	−1.85; 2.87
L1 (Parent Level)																		
parental PD							−0.3	−2.95; 3.54	1.0	−2.35; 4.26	0.7	−2.39; 3.78	−0.3	−3.48; 2.91	0.6	−2.76; 3.86	0.2	−2.79; 3.16
parent age													−0.3 *	−0.56; −0.09	−0.29 *	−0.53; −0.04	−0.3 *	−0.48; −0.03
parent sex													−2.5	−5.83; 0.93	−1.9	−5.44; 1.58	−1.9	−5.01; 1.29
Random Effect																		
L1 (child)	47.7 **	36.16; 63.03	66.0 **	50.03; 87.00	76.8 **	58.64; 100.5	47.8 **	36.17; 63.07	66.1 **	50.07; 87.13	76.8 **	58.67; 100.5	47.2 **	35.61; 62.66	57.6 **	43.17; 76.73	76.1 **	58.28; 99.36
L2 (family)	65.4 **	46.79; 91.43	56.7 **	37.08; 86.78	36.4 **	19.74; 67.08	65.4 **	46.73; 91.39	56.4 **	36.80; 86.53	36.2 **	19.62; 66.95	58.3 **	40.67; 83.56	58.1 **	38.60; 87.29	27.1 **	12.82; 57.41
ICC	0.5781		0.4623		0.3216		0.5778		0.4607		0.3207		0.5524		0.5024		0.2626	
−2 Log-Likelihood	2153.3		2185.9		2182.9		2153.3		2185.5		2182.7		2122.3		2148.5		2145.2	
$\chi^{2}/\mathrm{df}$							0.03/1		0.32/1		0.32/1		31.0/4		37.0/4		37.5/4	
*p*							>0.05		>0.05		>0.05		<0.01		<0.01		<0.01	
BIC	2170.4		2202.9		2199.9		2176.0		2208.2		2205.4		2167.6		2193.8		2190.5	

Note: Multilevel-model; N = 253 children nested in 171 families. CBCL-T: CBCL total problem score, CBCL-I: CBCL internalizing scale; CBCL-E: CBCL externalizing scale; PD: personality disorder; ICC: Intraclass correlation coefficient; BIC: Bayesian Information Criterion; * *p* ≤ 0.05; ** *p* ≤ 0.01.

**Table 6 children-09-01697-t006:** BSI GSI cut-off (mentally ill parent) as a predictor for CBCL (children’s psychiatric symptoms).

	Model 0						Model 1						Model 2					
	CBCL-T	CI-95%	CBCL-I	CI-95%	CBCL-E	CI-95%	CBCL-T	CI-95%	CBCL-I	CI-95%	CBCL-E	CI-95%	CBCL-T	CI-95%	CBCL-I	CI-95%	CBCL-E	CI-95%
Fixed Effects																		
Intercept	59.4 **	57.62; 61.24	59.4 **	49.82; 94.62	56.7 **	55.01; 58.42	57.3 **	55.31; 59.23	57.3 **	55.33; 59.33	55.2 **	53.28; 57.07	65.7 **	52.48; 79.00	53.6 **	39.69; 67.49	70.51**	57.64; 83.37
L1 (Children Level)																		
child age													0.2	−0.21; 0.53	0.6 **	0.20; 1.01	−0.3	−0.71; 0.10
child sex													0.4	−2.21; 3.01	2.4	−0.46; 5.28	−1.6	−4.49; 1.22
L1 (Parent Level)																		
parent BSI GSI cut-off							8.3 **	4.41; 12.14	7.9 **	4.02; 11.86	5.7 **	2.03; 9.45	7.7 **	3.75; 11.70	8.2 **	4.13; 12.36	4.80 *	1.13; 8.48
parent age													−0.3 **	−0.64; −0.52	−0.2	−0.49; 0.12	−0.4 *	−0.64; −0.07
parent sex													2.2	−1.40; 5.70	1.0	−2.69; 4.68	3.0	−0.30; 6.31
Random Effect																		
L1 (child)	47.7 **	34.54; 65.95	68.7 **	49.82; 94.62	77.1 **	56.60; 104.9	48.3 **	34.86; 67.03	69.2 **	50.18; 95.51	79.3 **	57.80; 108.8	48.1 **	34.68; 66.69	61.9 **	44.68; 85.78	76.0 **	55.35; 104.4
L2 (family)	74.2 **	50.49; 108.9	62.2 **	38.06; 101.5	42.7 **	22.76; 80.20	59.9 **	38.99; 92.10	49.2 **	27.84; 86.83	33.4 *	15.08; 74.01	56.9 **	32.64; 88.49	54.3 **	32.53; 90.56	27.4 *	11.12; 67.38
ICC	0.6085		0.4751		0.3567		0.5535		0.4153		0.2964		0.5421		0.4671		0.2647	
−2 Log-Likelihood	1508.0		1530.4		1529.2		1491.3		1515.4		1520.4		1471.6		1492.1		1489.5	
$\chi^{2}/\mathrm{df}$							16.7/1		15.0/1		8.8/1		19.8/4		23.3/4		30.8/4	
*p*							<0.01		<0.01		<0.01		<0.01		<0.01		<0.01	
BIC	1524.0		1546.3		1545.1		1512.6		1536.6		1541.6		1513.9		1534.4		1531.9	

Note: Multilevel-model; N = 203 children nested in 134 families. CBCL-T: CBCL total problem score, CBCL-I: CBCL internalizing scale; CBCL-E: CBCL externalizing scale; BSI GSI: Brief Symptom Inventory Global Severity Index; ICC: Intraclass correlation coefficient; BIC: Bayesian Information Criterion; * *p* ≤ 0.05; ** *p* ≤ 0.01.

## Data Availability

Data available on request from the authors.

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
