# Peer review of "Clinical Trial Data: Both Parents Having Psychiatric Symptoms as Risk Factor for Children’s Mental Illness"

_children, 2022, doi:10.3390/children9111697_

Round 1

Reviewer 1 Report

The authors present a study designed to assess the relationship between a parent's mental health and their child's mental health.  As the author's note, this is a well-studied area, and it is not entirely clear what this work adds to the field.

Major Points:
1. The manuscript is sloppy.  There are numerous English-language errors and typos.  The phrase is "fright without solution," not freight.  Likert is a name and should be capitalized.  Abbreviations are used in the Abstract and elsewhere without prior definition.  The references are not in MDPI format for the journal.  Etc.

2. I have read the first two paragraphs on page 4 numerous times, and I still cannot make sense of the numbers and percentages.  Given the family situation, the children could be exposed to a single parent or as many as four (divorced and remarried / partnered).  It is unclear how many individuals fall into each category.  And in the case of a single-parent home, there is still the biological contribution of the other parent to the child to consider.

3. The DSM-5 does not use the Axis system, and it has been in place for almost a decade.  What is the justification for using DSM-IV criteria?

4. The impact of co-morbities is not emphasized enough in my view.  This is especially true for co-morbities than represent different axes.

5. I am not very familiar with the statistical approach used here, so I defer to other reviewers on that point.

Reviewer 2 Report

First, I would like to congratulate the Authors on their great study. The manuscript entitled "Clinical Trial Data: Both Parents Having Psychiatric Symptoms as Risk Factor for Children’s Mental Health" are a valuable work in this topic. The introduction part is well-detailed. It gives to the reader a comprehensive view of the research problem and the research questions as well. The methods are detailed which makes the results clear. In the last part, the Authors conclude relevant things from their results. Even though, It is a great manuscript I still would like to add some suggestions to the Authors to improve their study even more.

My comments especially on the introduction and the methods parts:

·       Please define mental health and mental disorders. 

·       Please add hypotheses to the end of the introduction.

·       A minor, but relevant thing is the citation in the text is incorrect (We do not use dates in this format, e.g., line 109).

·       How many items were used for each measure? It seems there is no information on that.

Round 2

Reviewer 1 Report

Much improved manuscript, and the letter addressed my concerns sufficiently.  The only typo I saw was the sample size in line 171.  To be consistent with the other numbers, I think that n should be 134, not 131.